# ML-TOMCAT: Machine-Learning-Based Satellite-Corrected Global Stratospheric Ozone Profile Data Set from a Chemical Transport Model

Sandip S. Dhomse[1,2], Carlo Arosio[3], Wuhu Feng[1,4], Alexei Rozanov[3], Mark Weber[3], and Martyn P. Chipperfield[1,2]

[1]School of Earth and Environment, University of Leeds, Leeds, UK
[2]National Centre for Earth Observation, University of Leeds, Leeds, UK
[3]Institute for Environmental Physics, University of Bremen, Bremen, Germany
[4]National Centre for Atmospheric Science, University of Leeds, Leeds, UK

**Correspondence:** Sandip S. Dhomse (s.s.dhomse@leeds.ac.uk)

**Abstract.**

High quality stratospheric ozone profile data sets are a key requirement for accurate quantification and attribution of long-term ozone changes. Satellite instruments provide stratospheric ozone profile measurements over typical mission durations of 5-15 years. Various methodologies have then been applied to merge and homogenise the different satellite data in order to create longer term observation-based ozone profile data sets with minimal data gaps. However, individual satellite instruments use different measurement methods, sampling patterns and retrieval algorithms which complicate the merging of these different data sets. In contrast, atmospheric chemical models can produce chemically consistent long-term ozone simulations based on specified changes in external forcings, but they are subject to the deficiencies associated with incomplete understanding of complex atmospheric processes and uncertain photochemical parameters.

Here, we use chemically self-consistent output from the TOMCAT 3-D chemical transport model (CTM) and a Random-Forest (RF) ensemble learning method to create a merged 42-year (1979-2020) stratospheric ozone profile data set (ML-TOMCAT V1.0). The underlying CTM simulation was forced by meteorological reanalyses, specified trends in long-lived source gases, solar flux and aerosol variations. The RF is trained using the Stratospheric Water and OzOne Satellite Homogenized (SWOOSH) data set over the time periods of the Microwave Limb Sounder (MLS) from the Upper Atmosphere Research Satellite (UARS) (1991-1998) and Aura (2005-2016) missions. We find that ML-TOMCAT shows excellent agreement with available independent satellite-based data sets which use pressure as vertical coordinate (e.g. GOZCARDS, SWOOSH for non-MLS periods) but weaker agreement with the data sets which are altitude-based (e.g. SAGE–CCI–OMPS, SCIAMACHY-OMPS). We find that at almost all stratospheric levels ML-TOMCAT ozone concentrations are well within uncertainties of the observational data sets. The ML-TOMCAT (V1.0) data set is ideally suited for the evaluation of chemical model ozone profiles from the tropopause to 0.1 hPa and is freely available via https://doi.org/10.5281/zenodo.5651194 (Dhomse et al., 2021).

# 1 Introduction

With the successful implementation of the Montreal Protocol, various observations confirm reductions in the concentrations of halogenated ozone-depleting substances (ODSs) in the atmosphere (WMO, 2014, 2018). Satellite data records also confirm a peak in upper stratospheric HCl (the main chlorine reservoir) around 1997, followed by a steady decline (Anderson et al., 2000; Froidevaux et al., 2006a; Hossaini et al., 2019). Hence, attention has turned towards the detection and attribution of ozone recovery (e.g. Dhomse et al., 2006; Solomon et al., 2016; Chipperfield et al., 2017; Steinbrecht et al., 2017; Dhomse et al., 2018; Szeląg et al., 2020). However, the accurate quantification of ozone changes is challenging because of the quality of long-term ozone profile data sets, where measurement errors are of similar or larger magnitude than the long-term ozone trends. In addition, complex coupling between various physical and chemical processes controlling stratospheric ozone concentrations cause large short-term ozone changes. Complications also arise because there are some non-linear changes in stratospheric dynamics as well as chemical constituents. For example, between 2018 and 2021, some of the largest and smallest ozone losses of the recent decades were recorded in both the Arctic and Antarctic polar stratospheres (e.g. Wargan et al., 2020; Wohltmann et al., 2020; Bognar et al., 2021; Weber et al., 2021). Some observational data suggest that there has been a continuous decline in lower stratospheric ozone (Ball et al., 2018, 2020), which could be attributed to changes in stratospheric dynamics (e.g. Chipperfield et al., 2018; Wargan et al., 2018; Orbe et al., 2020; Abalos and de la Cámara, 2020). Atmospheric concentrations of ODSs such as CFC-11 are decreasing at uneven rates (Montzka et al., 2018, 2021) which could induce variability in ozone trends. Additionally, significant positive trends have been detected in very short-lived substances (VSLS) containing chlorine and bromine that are not controlled by the Montreal Protocol (e.g. Hossaini et al., 2015, 2019).

As there are no long-term ozone profile data from a single satellite instrument, various attempts have been made to merge such data from different instruments. However, individual satellite instruments have different temporal and spatial resolution depending on the measurement techniques and retrieval algorithms (e.g. Sofieva et al., 2014; Damadeo et al., 2018). For example, solar occultation instruments (e.g. Stratospheric Aerosol and Gas Experiment (SAGE, McCormick et al., 1989), Halogen Occultation Experiment (HALOE, Russell III et al., 1993)) provide high quality measurements but are constrained by limited spatial coverage. Limb-scanning instruments such as the Microwave Limb Sounder (MLS, Froidevaux et al., 2006b), Scanning Imaging Absorption Spectrometer for Atmospheric Cartography (SCIAMACHY, Bovensmann et al., 1999), Optical Spectrograph and InfraRed Imager System (OSIRIS, Murtagh et al., 2002 ) provide better spatial coverage but have coarser vertical resolution. A key constraining factor is that only few satellite data sets cover enough overlapping years to remove inter-instrument biases with minimal uncertainty.

Hence, Randel and Wu (2007) adopted a novel approach to create a gap-free stratospheric ozone profile data for the 1979–2005 time period. They used SAGE (I and II) satellite profile measurements and polar ozonesondes, together with a seasonally varying ozone climatology from Paul et al. (1998) to fill the gaps, to generate multi-variate regression-based gap-free ozone anomalies. Later, Cionni et al. (2011) used a similar methodology along with climate model simulations to extend the time series backwards to 1850. The Cionni et al. (2011) data were recommended for the historical CMIP5 simulations for the climate models that did not include stratospheric chemistry, in order to enforce time-dependent ozone variations. Hassler

et al. (2008) used a different methodology to create a satellite-based long-term ozone profile data set. Along with SAGE I and II measurements, they used HALOE and POAM (Polar Ozone and Aerosol Measurement) II and III satellite measurements, as well as ozonesonde data from 130 stations, to create a collection of binary data files; also known as the "Binary DataBase of Profiles" (BDBP) version 1.0. Bodeker et al. (2013) updated the BDBP data set to construct "Bodeker Scientific" or "BS" data. They updated BDBP data by including measurements from the Limb Infrared Monitor of the Stratosphere (LIMS), the Improved Limb Array Spectrometer (ILAS), and ILAS II. They used a multivariate regression model to create different versions of the ozone profile data set ranging from the surface to 70 km for the 1979-2008 time period. Hassler et al. (2018a) revised and extended (1979–2016) the "BS" data set by using the TOMCAT chemical transport model (CTM) ozone profiles as a transfer function to capture ozone variability for the period without satellite observations.

Another widely used merged data set is the Global OZone Chemistry And Related trace gas Data records for the Stratosphere (GOZCARDS, Froidevaux et al., 2015). These are monthly mean zonally averaged time series constructed using ozone profile measurements from several NASA satellite instruments and the Atmospheric Chemistry Experiment - Fourier Transform Spectrometer (ACE-FTS, Bernath et al., 2005). Merging is done primarily by removing average biases between SAGE II and individual data records for overlap periods (Froidevaux et al., 2015). The GOZCARDS data files contain mixing ratios on a pressure–latitude grid (316 hPa to 0.1 hPa), updated later to GOZCARDS v2.2 (Froidevaux et al., 2019).

Davis et al. (2016) adopted a slightly different approach to construct the Stratospheric Water and Ozone Satellite Homogenized (SWOOSH) data set. SWOOSH merges stratospheric ozone profile data from solar occultation instruments (SAGE-II/III, HALOE, ACE-FTS) as well as limb-scanning instruments (UARS-MLS, and Aura-MLS). The measurements are homogenized by applying corrections that are calculated from data taken during time periods of instrument overlap. The primary SWOOSH data product consists of monthly mean zonal-mean values on a pressure grid at 2.5, 5 and 10 degree resolution. One of the major characteristics of SWOOSH data is that when merging greater weight is given to the instruments that sample more frequently (e.g. Aura-MLS). Filled and unfilled versions of the data set exist on both geographical and equivalent latitude coordinates.

Several additional attempts have been made to merge satellite time series from limb and occultation instruments. For example, the SAGE–CCI–OMPS data set, described in Sofieva et al. (2017) includes SAGE II time series and several limb data sets. The OMPS-LP data set used is produced at the University of Saskatoon (Zawada et al., 2018). First, they screened and homogenized CCI data sets in the HARMOZ format before merging them in terms of ozone anomalies. Recently, Arosio et al. (2019) created a merged SCIAMACHY-OMPS limb data set (SCIA-OMPS), which combines these two time series produced at the University of Bremen. They used MLS data series as a transfer function to merge SCIAMACHY with OMPS-LP as these instruments share only two months of overlap, but MLS was not included in the merged data set. This time series is monthly averaged, covers the period 2002-present and is longitudinally resolved, with a $5°$ latitude $\times$ $20°$ longitude grid. Due to the similarities in the measurement geometries and techniques, and in the retrieval approaches, they implemented a plain de-biasing approach for the merging, directly obtaining a long-term ozone time series in appropriate units.

Another widely accepted approach is using data assimilation techniques to create observation-based data (e.g. Feng et al., 2008; Skachko et al., 2014; Errera et al., 2019; Wargan et al., 2020). However, only a few instruments such as MLS provide relatively long-term ozone profile measurements. For the pre-MLS time period, very few observations are available that can

provide good constraint on the assimilation system. Also, the forward model is generally forced with available (re)analysis dynamical fields so reanalysis data sets are also prone to the inhomogenities in the forcing fields along with any discrepancies in chemical scheme.

In this paper we present a new data-model method for producing a long-term data set of stratospheric ozone. We use ozone profile output from a CTM to create a machine-learning-based satellite-corrected long-term chemically (and dynamically) consistent ozone profile data set (hereafter, ML-TOMCAT) for the 1979 – 2020 time period. The CTM setup is described in Section 2, followed by our methodology in Section 3. Comparisons of ML-TOMCAT with some of the other available merged ozone profile data sets are presented in Section 4, with a summary of our key results in Section 5.

## 2   Model Setup

We use chemically consistent monthly mean zonal mean ozone profiles from the TOMCAT CTM as the basis data set. TOM-CAT is an off-line three-dimensional (3D) CTM that includes a comprehensive stratospheric chemistry scheme (Chipperfield, 2006). For the present study, the CTM setup is similar to the control simulations used in our recent studies such as Feng et al. (2021); Bognar et al. (2021) and Weber et al. (2021). Briefly, TOMCAT is forced with meteorological fields from ERA-5 reanalyses (Hersbach et al., 2020), starting from 1979. Simulations are performed at a $2.8 \times 2.8$ degree horizontal resolution with 32 hybrid sigma-pressure levels extending from the surface to about 60 km. For major ODSs and GHGs the model uses time-dependent observed global mean surface mixing ratios (Carpenter et al., 2018) that are treated as well-mixed throughout the troposphere. The model also includes the effects of solar flux variability and heterogeneous chemistry on volcanically enhanced stratospheric aerosol as described in Dhomse et al. (2015, 2016). Solar irradiance data are from the NRL2 (Coddington et al., 2016) empirical model and the sulfate aerosol surface area density (SAD) from Luo (2016). TOMCAT also includes chlorine and bromine contributions from VSLS as described in Hossaini et al. (2019). A passive ozone tracer (no chemical ozone loss), generally used to diagnose chemical ozone loss, is initialised every six months from the chemical ozone tracer (1st June and 1st December). TOMCAT has been regularly used to study long-term changes in stratospheric trace gases, showing good agreement with various ground-based and satellite data sets (e.g. Mahieu et al., 2014; Chipperfield et al., 2015; Wales et al., 2018; Harrison et al., 2021; Prignon et al., 2021).

## 3   Methodology

We use the Random Forest (RF) regression analysis to generate a long-term chemically consistent data set. The RF is a supervised machine learning (ML) algorithm that uses an ensemble of decision trees (e.g. Breiman, 2001; Svetnik et al., 2003). A decision tree can be considered as a flow chart used in computer programming (a tree-shaped schematic) that is generally used to show a statistical probability or path of action. A single decision tree in a RF can be considered as a random tree in a forest of decision trees. Each decision tree consists of three components: decision nodes, leaf nodes and a root node. The root

node and decision nodes of the decision tree represent the explanatory variables. The leaf nodes represent the final output. The explanatory variables used in our analysis are explained at end of this section.

A decision tree algorithm divides the data set into branches (using true and false criteria), which further segregate into other branches until a leaf node (or result node) is reached. Multiple trees are constructed by randomly sampling data points multiple times (e.g. bootstrap method). Hence, an individual tree can be considered as a unique tree (hence unique output). RF uses a

125 bagging technique, that means the RF model consists of many individual decision trees and aggregated predictions are used for the final prognosis. A distinct advantage of RF regression is that it is relatively accurate and very easy to set up. RF can also behave like a non-linear regression method. As RF adds randomness to the decision procedure, instead of relying on the most important explanatory variables, it searches for the best variable among random subsets. This ensures that the final output does not rely heavily on a single explanatory variable, thereby avoiding over-fitting (e.g. Kotsiantis, 2013). We use RF Regression

from the Python package Sci-kit learn (Pedregosa et al., 2011) with two options: *random_state=0*, and *bootstrap=True*.

Initially, TOMCAT zonal mean ozone profiles are linearly interpolated in log-pressure space on to 43 equidistant (12 per decade) pressure levels (1000–0.1 hPa, MLS pressure levels), followed by spatial interpolation onto 72 SWOOSH latitude bins at 2.5° resolution. SWOOSH data are obtained via https://csl.noaa.gov/groups/csl8/swoosh/. Then, we calculate the ozone difference ($dO_3$) between SWOOSH and model ozone profiles for the 1991-1998 and 2005-2016 time periods (total 20 years).

For the calculation of $dO_3$ values, we use the gap-filled SWOOSH data product. SWOOSH data ranges from 316 to 1 hPa (31 pressure levels), hence for pressure levels below 316 hPa the ML-TOMCAT values are set to latitudinally and monthly varying climatological values from Logan (1999) which are also used in stratospheric TOMCAT simulations.

For the regression analysis, a 20-year (largely MLS covering) time period is selected in order to avoid heteroscedasticity (i.e. effect of different sampling frequencies/methodologies (e.g. Sofieva et al., 2014; Millán et al., 2016) between different types

of satellite data sets) as SWOOSH relies heavily on MLS (UARS and Aura) data records. Additionally, it also covers a period when the stratospheric chlorine loading was increasing (1991-1998) and decreasing (2005-2016) and RF has enough sample to include different characteristics of ozone variability. The regression model has 5 terms: Passive ozone ($PO_3$), HCl mixing ratio ($HCl$), methane mixing ratio ($CH_4$) as well as observation-model total column difference ($dTCO$) and Mg II solar flux term ($MgII$). The $PO_3$, $HCl$ and $CH_4$ terms account for possible biases in CTM profiles due to transport in different stratospheric

regions (e.g. Strahan et al., 2011; Feng et al., 2021). $dTCO$ is an ideal learner for the lower stratospheric ozone transport as total column ozone measurements have much smaller retrieval errors (e.g. Petropavlovskikh et al., 2019), hence they provide a good constraint for the possible biases in ERA-5 stratospheric transport (e.g. Ploeger et al., 2021). TOMCAT has 203 spectral bins in the photolysis scheme (e.g. Dhomse et al., 2016). Hence, the $MgII$ solar flux term is included to account for possible biases in the representation of the 11-year solar flux variability (e.g. Haigh et al., 2010; Dhomse et al., 2013) or the use of

coarse spectral bins (e.g. Sukhodolov et al., 2016).

Overall, there are five explanatory variables in the regression model for individual grid points and these are taken from TOMCAT output fields. The regression model can be represented as:

$$dO_3 = \beta_1 PO_3 + \beta_2 HCl + \beta_3 CH_4 + \beta_4 dTCO + \beta_5 MgII + residuals \tag{1}$$

where $\beta_1$, $\beta_2$, $\beta_3$, $\beta_4$ and $\beta_5$ can be considered as the contribution coefficient for a given explanatory variable and $PO_3$, $HCl$, $CH_4$ are TOMCAT monthly mean zonal mean tracers. For the calculation of $dTCO$ we use Copernicus Climate Change Service (C3S) total ozone data (1979–2018). The C3S total column product is a combination of total column data from 15 sensors using gap-filling assimilation methods and is obtained via https://cds.climate.copernicus.eu/cdsapp{#!}/dataset/satellite-ozone?tab=overview (last access: 1 May 2021). For the years 2019 and 2020, we use level 3 total column data from the Ozone Monitoring instrument (OMI) V3 that is obtained via https://search.earthdata.nasa.gov (last access: 1 May 2021). The Mg II index is obtained from http://www.iup.uni-bremen.de/UVSAT/datasets/mgii (last access: 1 May 2021). We assume long-term chemical ozone changes are realistically represented by TOMCAT chemistry (e.g. Feng et al., 2007; Chipperfield et al., 2017; Dhomse et al., 2019), hence all the predictor time series are detrended and normalised between 0 to 1.

## 4 Results

Atmospheric chemical models are ideal tools for understanding/simulating past (and future) ozone changes, as they combine up-to-date knowledge about various physical and chemical processes using a mathematically consistent framework. Different models use different combinations of chemical and dynamical schemes to represent important processes in the atmosphere. However, some of these processes are computationally expensive, hence they are represented by somewhat simplified parameterisations. For example, many chemical models prescribe observation-based sulfate surface area density (SAD) to represent the effects of volcanically enhanced stratospheric aerosol for simulating heterogeneous chemistry which leads to ozone loss (Dhomse et al., 2015). Many models also prescribe surface concentrations of greenhouse gases (GHGs) and ODSs rather than emission fluxes. CTMs such as TOMCAT use dynamical forcing fields from (re)analyses data sets such as ERA-Interim or ERA-5. Hence CTMs are subject to possible inhomogeneities due to changes in the number of assimilated observations, as well as other deficiencies (e.g. missing processes) in the forward model used in the assimilation system. On the other hand, observational data sets are also subject to errors associated with the measurement techniques, instrument degradation and retrieval algorithms. Hence, almost all chemical models may be expected to show a bias against observational data records, either because of model deficiencies or errors in the observations. However, chemical models do use a consistent chemical scheme, so we can assume that chemical model-observation ozone differences are largely due to uncertainties in the forcing fields such as meteorology (e.g. winds, temperature) and chemical parameterisations (e.g. reaction rates, solar fluxes, photolysis schemes). CTMs have the distinct advantage in terms of dynamics as they are forced with up-to-date reanalysis data, although with the above-noted caveat of possible inhomogeneities in observations used in the assimilation systems. Hence, recently initiated SPARC Reanalysis Intercomparison project (S-RIP) is aimed at providing guidance on future reanalysis activity. S-RIP also plans to perform comprehensive evaluation and intercomparison of different reanalysis data products; for details see https://www.sparc-climate.org/sparc-report-no-10. Here, we train the ML algorithm on the model-observation differences for the period that has relatively good temporal sampling. Estimated parameters are then used to simulate differences for the entire (1979–2020) time period. In this section, we analyse model-observation biases associated with individual predictors and compare the ML-corrected data against a variety of observation-based data sets.

## 4.1 Model biases

Figure 1 shows climatological (2006–2020) monthly zonal mean differences between TOMCAT and SWOOSH ozone profiles (TOMCAT minus SWOOSH). TOMCAT profiles show an almost symmetrically structured negative biases in the upper stratosphere and positive biases in the lower stratosphere. The largest negative biases (up to 0.8 ppm) occur in the tropical upper stratosphere (around 3 hPa) and they remain negative throughout the year. The ozone lifetime at these altitudes is less than a day, hence the observed biases might be associated with deficiencies in the photochemical reactions in the model. At this altitude, ozone production is largely controlled by solar fluxes below 240 nm while longer wavelengths control ozone destruction (e.g. Haigh et al., 2010). Therefore, negative ozone biases in the upper stratosphere are most probably due to uncertainties in the solar irradiances and/or photolysis cross sections that control ozone production (e.g. Brasseur and Solomon, 2006). Furthermore, in this region of the atmosphere, ozone chemistry is mostly temperature dependent (e.g. Stolarski et al., 2010; Dhomse et al., 2013, 2016), hence the model ozone biases could be due to uncertainties in temperature-dependent reaction rates (e.g. Ghosh et al., 1997).

In the lower stratosphere the ozone lifetime ranges from months to years, hence positive biases in the TOMCAT ozone could be due to a combination of both dynamics and chemistry. First, reduced overhead ozone could increase lower stratospheric ozone via the self-healing effect, i.e. increased ultra-violet radiation increases ozone production at lower altitudes (e.g. Haigh, 1994). Second, ozone is primarily produced in the tropical stratosphere, and its downward transport is controlled by the quasi-biennial oscillation (QBO) (e.g. Tian et al., 2006), whereas transport towards mid-high latitudes is determined by the strength of the Brewer-Dobson (BD) circulation (e.g. Holton et al., 1995; Weber et al., 2003; Dhomse et al., 2006; Weber et al., 2011) which increases its lifetime considerably. Hence, ozone biases in the lower stratosphere are likely due to the incomplete representation of various circulation pathways in TOMCAT either due to model resolution or missing representation of key physical process in the ERA-5 reanalysis scheme (e.g. Mitchell et al., 2020) which impacts the meteorology used in the CTM.

## 4.2 Contribution from explanatory variables

As the exact causes of TOMCAT ozone biases are still not well understood, we use the RF model to remove them. The RF regression model coefficients are derived using 20 years (1991–1998, 2006–2018) for which SWOOSH data includes a large number of observational profiles especially from MLS on the UARS and Aura satellite platforms. The RF regression model uses 20 years of monthly data with 80% and 20% of data points being used for training and testing, respectively. The estimated RF regression coefficients are then used to calculate model biases for the entire 42-year time period (1979–2020). RF-calculated ozone biases are then added to the TOMCAT time series to create the long-term gap-free data set, hereafter labelled ML-TOMCAT.

Figure 2 shows how much variance (or $R^2$) of the data the RF regression model is able to explain, along with regression coefficients for individual explanatory variables. For example, $R^2$ value of 0.8 indicates that the RF regression model is able to explain 80% of the biases in TOMCAT ozone relative to SWOOSH data for the 20 years of the training period. $R^2$ also represents sum of the regression coefficients from individual explanatory variables. Overall, the RF regression model performance

is consistently high ($R^2 > 0.8$) throughout the stratosphere, except for the mid-stratosphere which is a transition region where the TOMCAT ozone biases change from positive to negative. At high northern latitudes, mid-stratospheric $R^2$ values decrease to 0.6. However, since TOMCAT – SWOOSH differences are much smaller here, a RF-based correction has a minimal impact on the quality of ML-TOMCAT ozone profiles.

Additionally, as expected, the RF regression coefficients are significant in different regions of the stratosphere for various explanatory variables. The passive ozone tracer seems to show the largest coefficients in the tropical mid-stratosphere, as well as varying contributions in different regions of the stratosphere. The passive ozone contribution in the tropical mid-stratospheric could be linked to the incomplete representation of $NO_x$-related chemical changes in TOMCAT and/or seasonal changes in the stratospheric transport in the re-analysis (e.g. Galytska et al., 2019). The $HCl$ tracer shows significant coefficients in the upper stratosphere, where the ClO ozone loss cycle is important. It also shows significant contribution at low-mid-latitude lower stratosphere. $HCl$ can be considered as both a dynamical and chemical proxy, as in the upper stratospheric $HCl$ is primarily produced via degradation of ozone-depleting substances and is transported downwards at high latitudes via the BD circulation (e.g. Mahieu et al., 2014). Therefore, $HCl$ variations in this region can be considered as a proxy for the changes in the strength of the BD circulation as well as horizontal isentropic transport, especially between tropics and mid-latitudes. The $CH_4$ tracer term seems to show significant coefficients in the lowermost stratosphere (just above the tropopause) as well as a significant contribution around the mid-latitude sub-tropics. The $CH_4$ tracer contribution resembles a QBO-induced secondary circulation pattern. Interestingly, the solar term shows the largest coefficients in the mid-latitude upper stratosphere rather than in the tropical upper stratosphere, suggesting solar flux variability has only a minor contribution to the TOMCAT-observation biases. As expected the $dTCO$ term shows the largest contribution in the lowermost stratosphere, especially in the tropical and polar regions. Interestingly, ozone anomalies in these regions show good agreement with various satellite-based data sets (e.g. Chipperfield et al., 2017, 2018; Li et al., 2020; Feng et al., 2021), and TOMCAT biases are much smaller. This means that although $dTCO$ coefficients are largest in the lowermost stratosphere, the overall bias correction contribution remains relatively small.

## 4.3  Comparison against merged data sets

After analysing the regression coefficients, we now present a comparison between ML-TOMCAT and available satellite-based long-term data sets. Due to key differences between satellite measurement techniques, ozone profiles are retrieved either at altitude or pressure levels and either in units of mixing ratio or number density. For example, MLS retrieves profiles of ozone mixing ratio on pressure levels whereas SAGE retrieves profiles of number density on altitude levels. Hence, merging these different data sets needs pressure, temperature or altitude information at a given co-location from an external source such as reanalysis data. The GOZCARDS and SWOOSH data sets use MERRA2 reanalysis data to convert SAGE II ozone number density profiles on fixed pressure levels (Damadeo et al., 2013). ML-TOMCAT is based on modelled ozone profiles as a function of pressure, although conversion to altitude (geopotential height) coordinates is straightforward. In particular, ML-TOMCAT data were processed on corresponding grids/units using ERA-5 geopotential height, temperature and pressure fields that are used to drive TOMCAT.

This subsection consists of two parts. First we compare ML-TOMCAT profiles with data sets using pressure co-ordinate systems (e.g. SWOOSH, GOZCARDS), followed by comparisons with altitude-based data sets (SAGE–CCI–OMPS, SCIA-OMPS, BSVert).

### 4.3.1 Comparison with pressure level data

As noted earlier, we used only 20 years of SWOOSH data to train the RF model. Hence, the next obvious step is to compare ML-TOMCAT ozone with SWOOSH over the full time period. Figure 3 compares relative differences (in percent) of ML-TOMCAT with GOZCARDS and SWOOSH, respectively, as a function of latitude and pressure. ML-TOMCAT shows slightly positive biases in the middle stratosphere and somewhat negative biases in the upper and lower stratosphere with respect to both SWOOSH and GOZCARDS data. The largest biases (up to 10%) are observed in the tropical lowermost stratosphere as well as polar latitudes. However, these largest differences in the tropical lowermost stratosphere (and upper troposphere) cannot be correctly validated as most satellite retrievals show largest their uncertainties in this region (Rahpoe et al., 2015; Steinbrecht et al., 2017; Sofieva et al., 2021). Similarly, for the non-MLS period, the biases in the polar stratosphere could be due to the lack of observational ozone profiles during polar night.

Figure 4 shows TOMCAT, ML-TOMCAT, SWOOSH and GOZCARDS ozone time series over the equator ($0°$ lat) at 3 pressure levels (1, 10 and 50 hPa). Supplementary Figures S1 to S10 show similar comparisons at $15°$N, $15°$S, $30°$N, $30°$S, $45°$N, $45°$S, $60°$N, $60°$S, $75°$N, and $75°$S latitude bins. The grey shaded area indicates the standard deviation of the ozone values within each bin for the GOZCARDS time series. The green shaded areas indicates the root mean square uncertainty of the combined data sets for each bin in SWOOSH data ($\sigma_{rmss}$ in Davis et al. (2016)). Overall, there is a good agreement between the ML-TOMCAT, GOZCARDS and SWOOSH time series. As seen in Figure 1, ML-TOMCAT shows significant improvements in the tropical stratosphere compared to TOMCAT.

A peculiar detail of Figure 4 is that the standard deviation in the SWOOSH time series is largest during the 1991-1999 time period, which could be due to a combination of various factors. First, UARS MLS ozone profiles are retrieved at only six levels per pressure decade (Livesey et al., 2003) instead of 12 levels per decade for Aura MLS (see https://mls.jpl.nasa.gov/data/v5-0_data_quality_document.pdf). Second, significant enhancement in the stratospheric aerosol loading following the Mt. Pinatubo eruption in June 1991 led to larger retrieval errors. Even with those uncertainties in SWOOSH (and GOZCARDS), ML-TOMCAT is generally close to the satellite-based data sets for the entire time period and the agreement with satellite data is greatly improved in comparison to the original TOMCAT profile data. Supplementary Figures S1 to S10 also show an excellent agreement between ML-TOMCAT and the GOZCARDS/SWOOSH data sets for other latitude bands.

Next we scrutinise percentage differences between GOZCARDS and ML-TOMCAT on the same pressure levels. Figure 5 shows relative differences between TOMCAT, ML-TOMCAT and SWOOSH ozone time series with respect to GOZCARDS. As seen earlier, TOMCAT ozone shows up to 40% positive biases in the lower stratosphere and 10% negative biases in the upper stratosphere (also seen in Figure 1). In contrast, ML-TOMCAT biases are well below 5% at all levels. At 50 hPa, TOMCAT biases seems to follow QBO-type oscillations that are correctly removed in ML-TOMCAT. Similarly, at 1 hPa TOMCAT differences show some uneven variations that could be linked to the inhomogeneities in the ERA-5 dynamical fields that are

used to force TOMCAT. Furthermore, ML-TOMCAT differences show much smaller and almost linear biases at 1 hPa and lie well within the spread of GOZCARDS data.

Interestingly, although both GOZCARDS and SWOOSH are created by merging nearly identical data sets, there are differences between them which are largest for the 1984 to 2004 time period. This indicates that even slight differences in merging methodology leads to large differences in the merged data set. Although we use completely independent output from a CTM as a basis data set, GOZCARDS-ML-TOMCAT differences are within the expected discrepancy between GOZCARDS and SWOOSH data sets, especially at 10 and 50 hPa.

Another notable feature in Figure 5 is that at 50 hPa ML-TOMCAT shows largest differences during 2020, which could be associated with the biases in ERA-5 dynamics during that period. A TOMCAT sensitivity simulation forced with ECMWF operational analysis data shows better agreement with MLS ozone variation during this period (e.g. Chrysanthou et al., 2021). In addition, larger differences seen during 1984 (50 hPa), 1988 (10 hPa) and 1996-1999 (1 hPa) are most probably associated with SAGE II sampling issues and/or inhomogeneities in ERA-5 dynamical fields. However, a detailed analysis of these biases

is out of scope of this study and it needs further investigation.

### 4.3.2  Comparison with altitude level data

We now compare ML-TOMCAT ozone profiles against altitude-based merged satellite data sets. Figure 6 shows the relative differences between TOMCAT/ML-TOMCAT vs SAGE-CCI-OMPS (Sofieva et al., 2017), BSVert (Hassler et al., 2018a) as well as SCIA-OMPS (Arosio et al., 2019) data sets as a function of altitude and latitude. The top panels (a and b) compare the

mean relative differences between the SAGE-CCI-OMPS data set, TOMCAT and ML-TOMCAT, respectively. Here TOMCAT shows large positive biases (up to 20%) in the lowermost stratosphere and negative biases (up to 15%) in the upper stratosphere. On the other hand, ML-TOMCAT shows only ±10% biases throughout the stratosphere. Larger biases are seen in the Antarctic stratosphere that could be attributed to the limited observational ozone profiles used to construct the altitude-based merged satellite data products. Interestingly, ML-TOMCAT shows largest biases (up to 30%) w.r.t. the BSVert data set, though

TOMCAT profiles (forced with ERA-Interim) are used as transfer function while constructing BSVert (Hassler et al., 2018a). In addition, in the lowermost stratosphere, biases are negative in the tropics and SH mid-latitudes and positive in the NH mid-high latitudes. Hence, a contributing factor for these hemispherically asymmetric biases with respect to BSVert ozone profiles might be differences between ERA-Interim and ERA-5 reanalysis data (e.g. Ploeger et al., 2021) that are used to force these two data sets. The negative values in relative differences in the lower tropical stratosphere shown with respect to the SCIA-OMPS

data set in the fourth panel is systematic throughout the time series and is thought to be related to two factors. The first one is the rather coarse vertical grid (corresponding to SCIAMACHY vertical resolution of 3.3 km) which makes it sensitive to the interpolation onto the TOMCAT grid. The second is the difference in use of merging procedure implemented for SCIA-OMPS and SWOOSH, so that ML-TOMCAT, trained over the MLS period using SWOOSH, shows a negative bias w.r.t. SCIA-OMPS, which however does not show such bias w.r.t. MLS (Arosio et al., 2019).

Figure 7 compares TOMCAT and ML-TOMCAT profiles with the three altitude-based ozone data sets with a focus on the equator (0° latitude). Supplementary Figures S11 to S20 show similar comparisons for 15°N, 15°S, 30°N, 30°S, 45°N, 45°S,

60°N, 60°S, 75°N, and 75°S latitude bins. Figure 8 displays the respective relative differences with respect to the SAGE-CCI-OMPS data set which in this case is taken as a reference. In this way it is possible to evaluate the improvement introduced by applying the ML algorithm but also have an estimation of the discrepancies between different merged data sets, which is expected to be on the order of 5-10%. With respect to the comparison with the data sets on pressure vertical coordinate, the scatter between the time series is larger here, due to the larger variety of different satellites available to produce the merged products and the fact that they have not been used in the ML training.

At about 45km in the tropics the ML algorithm seems to over-correct the negative bias shown by TOMCAT, leading to generally higher ozone values with respect to the other data sets, especially in the first half of the time series. In the middle stratosphere we find the best agreement between SAGE-CCI-OMPS and ML-TOMCAT; here the expected discrepancies among the merged data sets are comparable to the differences observed between ML-TOMCAT and SAGE-CCI-OMPS. At the peak of the ozone number profile around 25 km, we notice generally lower values for ML-TOMCAT, on average by 5%. Similar biases are observed at mid-high latitude as well as seen in Supplementary Figures S11 to S20. The strong seasonal cycle seen in the TOMCAT difference with respect to the merged data sets is largely reduced by ML-TOMCAT at this altitude.

### 4.3.3   Polar regions

The use of ML-TOMCAT helps to fill the observational gaps especially in atmospheric regions with lack of observations and before the beginning of the 21st century, when satellite measurements were sparser. For example, polar regions during local winter cannot be observed by limb observations based on scattered sunlight. Instruments such as Aura MLS and the Sounding of the Atmosphere using Broadband Emission Radiometry (SABER, Rong et al., 2008) have generally been used to fill this gap over the last two decades. For chemical models, complexities are also associated with the denitrification and dehydration (or chlorine activation) schemes that determine heterogeneous ozone losses (Grooß et al., 2018). Though most of our earlier studies showed that TOMCAT is able simulate to polar ozone losses quite realistically (e.g. Feng et al., 2007; Chipperfield et al., 2015, 2017; Dhomse et al., 2019), some systematic biases in polar stratosphere were noted in Feng et al. (2021) and Weber et al. (2021). Figure 9 compares ozone at 18 km over the Arctic demonstrating the good agreement between ML-TOMCAT and MLS in this region for both local summer and winter seasons. The bottom panel shows the ozone sub-column over the Antarctic (poleward of 70°S latitude) integrated between 12 and 20 km for TOMCAT, ML-TOMCAT and MLS averaged over September-October months. The good agreement between MLS and ML-TOMCAT during the ozone hole period is observed for most of the years. ML-TOMCAT enables the reconstruction of the large ozone losses which occurred in the 1980s during a phase when ozone depleting substances were on a rapid rise before the implementation of the Montreal Protocol and their phase out.

### 4.3.4   Total column comparison

As noted above, total column measurements have relatively small retrieval errors and high temporal resolution and thus provide an important data set for assessing model performance. Hence, we compare total column ozone from ML-TOMCAT and TOMCAT with the SBUV merged ozone (MOD) data set (urlhttps://acd-ext.gsfc.nasa.gov/Data_services/merged/). Monthly

mean total ozone columns, which are calculated by integrating number density profiles, are shown in Figure 10 for six latitude bands. Supplementary Figure S22 compares tropospheric columns obtained by integrating profiles for ozone volume mixing ratios below 150 parts per billion (ppb). As expected, TOMCAT tropospheric columns show a constant, repeating seasonal cycle with smallest mean value (and amplitude) in the tropics (up to 13 DU) and largest mean values in the NH mid-latitudes (up to 23 DU). In contrast, ML-TOMCAT tropospheric columns shows much larger mean and amplitude for all the latitude bins (mean column of about 35 DU in the NH mid-latitudes). ML-TOMCAT tropospheric columns also show large short-term variations (e.g. year 1991 following Pinatubo eruption), suggesting that the ML-TOMCAT pressure range (316 hPa - 1 hPa) does affect calculation of the tropospheric column through inclusion of levels that extend above 316 hPa. Note that below the 316 hPa level, both TOMCAT and ML-TOMCAT profiles include monthly climatological values from Logan (1999). Hence, it is important to note that ML-TOMCAT lower-mid tropospheric values are not recommended for scientific studies. Supplementary Figure S23 shows latitude-altitude cross section of climatological (1979–2020) ozone differences between ML-TOMCAT and TOMCAT in Dobson Units. Figure S23 clearly shows that largest differences are in the upper troposphere/lower stratosphere.

As noted earlier, TOMCAT total column differences are relatively small for both Arctic (60°N–90°N) and Antarctic (60°S – 90°S) regions and Figure 10 clearly shows that the same is true for ML-TOMCAT total columns as well. For mid-latitudes (35° – 60°), TOMCAT shows biases of up to +20 DU biases (especially in the NH mid-lat) compared to observations whereas ML-TOMCAT shows differences of less than 10 DU. Interestingly, for the tropics (20°S – 20°N), TOMCAT shows negative biases until 2000 and slightly positive biases afterwards that are almost negligible in ML-TOMCAT time series. On the other hand, for the near-global average (60°S – 60°N), ML-TOMCAT biases remain positive until 1990 and are close to TOMCAT biases. After 2000 TOMCAT seems to show slightly increasing positive biases w.r.t. SBUV MOD data but ML-TOMCAT seems to show almost negligible biases without any apparent trend.

Overall ML-TOMCAT ozone profiles outside the 316 – 1 hPa range should be considered as (slightly modified) TOMCAT model profiles. However, total column (and tropospheric column shown in Supplementary Figure S22) comparisons suggest that vertically integrated (1000 hPa – 0.1 hPa) ML-TOMCAT profiles can provide a useful estimate which is better than TOM-CAT and also better than combining ML-TOMCAT stratospheric column with tropospheric column from other sources (noting that levels above 1 hPa have a negligible contribution to the column). Hence, for convenience we include both tropospheric and lower mesospheric ozone values in ML-TOMCAT data files (1000 hPa – 0.1 hPa) even though they are only based on values from the TOMCAT model outside of the pressure range 316 – 1 hPa. For future versions of ML-TOMCAT we aim to also correct tropospheric ozone profile biases using merged tropospheric ozone profile data sets described, for example, in the Tropospheric Ozone Assessment Report (TOAR).

## 5 Summary and Conclusions

Stratospheric ozone concentrations are affected by many short- and long-term processes, hence high quality ozone profile data sets are needed for accurate attribution studies. Though satellite instruments provide global measurements, due to their short

mission durations various merging methodologies have been adopted to create homogeneous and gap-free long-term ozone profile data sets. Individual merging methodologies have distinct advantages and disadvantages. Atmospheric chemical models are also able to simulate chemically consistent long-term data sets, but they are prone to the deficiencies associated with the simplified parameterisations and uncertain parameters.

Here we have used TOMCAT CTM ozone profiles and a Random Forest (RF) regression model to create gap-free ozone profile data set (ML-TOMCAT) for 1979-2020. The RF is applied to the ozone difference between the SWOOSH and TOM-CAT ozone profiles by selecting 20 years of MLS measurements (UARS-MLS and AURA-MLS) as a training period. RF show consistent performance throughout the stratosphere, except at high latitudes and the mid-latitude mid-stratosphere. Overall, ML-TOMCAT shows excellent agreement with SWOOSH for the entire time period (1984–2020), though somewhat larger differences are apparent for the period where limited ozone measurements are available for SWOOSH construction. We also find that ML-TOMCAT shows better agreement with satellite-based merged data sets which use pressure as the vertical coordinate (e.g. SWOOSH, GOZCARDS) but weaker agreement with the data sets which use altitude (e.g. SAGE–CCI–OMPS, SCIA-OMPS). We find that at almost all stratospheric levels ML-TOMCAT ozone concentrations are well within uncertainties of the observational data sets. Presently, the ML-TOMCAT V1.0 data set is ideally suited for the evaluation of chemical model ozone profiles from the tropopause to 0.1 hPa. ML-TOMCAT V1.0 ozone profile data on pressure and altitude levels in mixing ratios and number density units is publicly available via. https://doi.org/10.5281/zenodo.5651194.

*Data availability.* We thank Sean Davies for SWOOSH data which is publicly available via https://csl.noaa.gov/groups/csl8/swoosh/. We also thank Lucien Froidevaux (Lucien.Froidevaux@jpl.nasa.gov) for GOZCARDS v2 data. SAGE–CCI–OMPS was obtained via http://www.esa-ozone-cci.org. SCIA-OMPS data is available upon email request to AR or MW. BSVert data were obtained fromhttps://zenodo.org/record/1217184 (Hassler et al., 2018b). ML-TOMCAT v1.0 data is publicly available via https://doi.org/10.5281/zenodo.5651194 (Dhomse et al., 2021)

.

*Acknowledgements.* SSD was supported by the NERC SISLAC project (NE/R001782/1). The financial support of part of this work from the State of Bremen, DAAD grant (CA), and ESA SOLVE Living Planet Fellowship (CA) is gratefully acknowledged. The UB contribution also benefited from the DFG VolImpact and BMBF SynopSys Projects. We thank NASA, NOAA and ESA for GOZCARDS, SWOOSH and SAGE-CCI-OMPS data products. We thank the European Centre for Medium-Range Weather Forecasts for providing their analyses. TOMCAT simulations were performed on the UK national Archer and Leeds Arc4 HPC systems.

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

*Author contributions.* SSD conceived the idea and initiated the study in discussion with MPC. The model runs were performed and analysed by SSD, MPC and WF. The figures were prepared by CA and SSD. The paper was written by SSD, MPC and MW who included comments from all of the other coauthors.

*Competing interests.* The authors have no competing interests

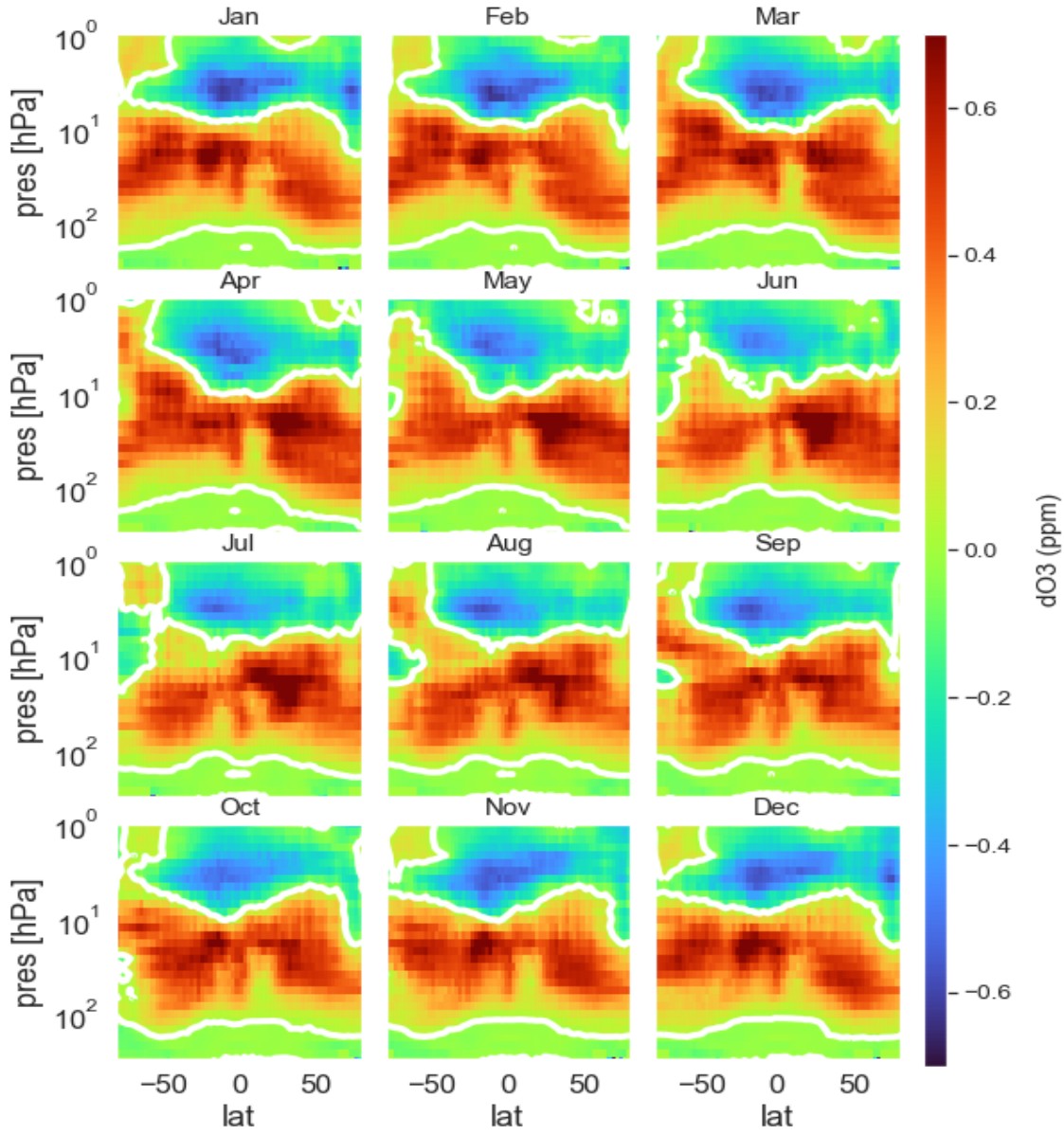

**Figure 1.** Latitude-pressure cross sections of the climatological (2006-2020) monthly mean difference (ppm) between TOMCAT and SWOOSH (Davis et al., 2016) ozone profiles.

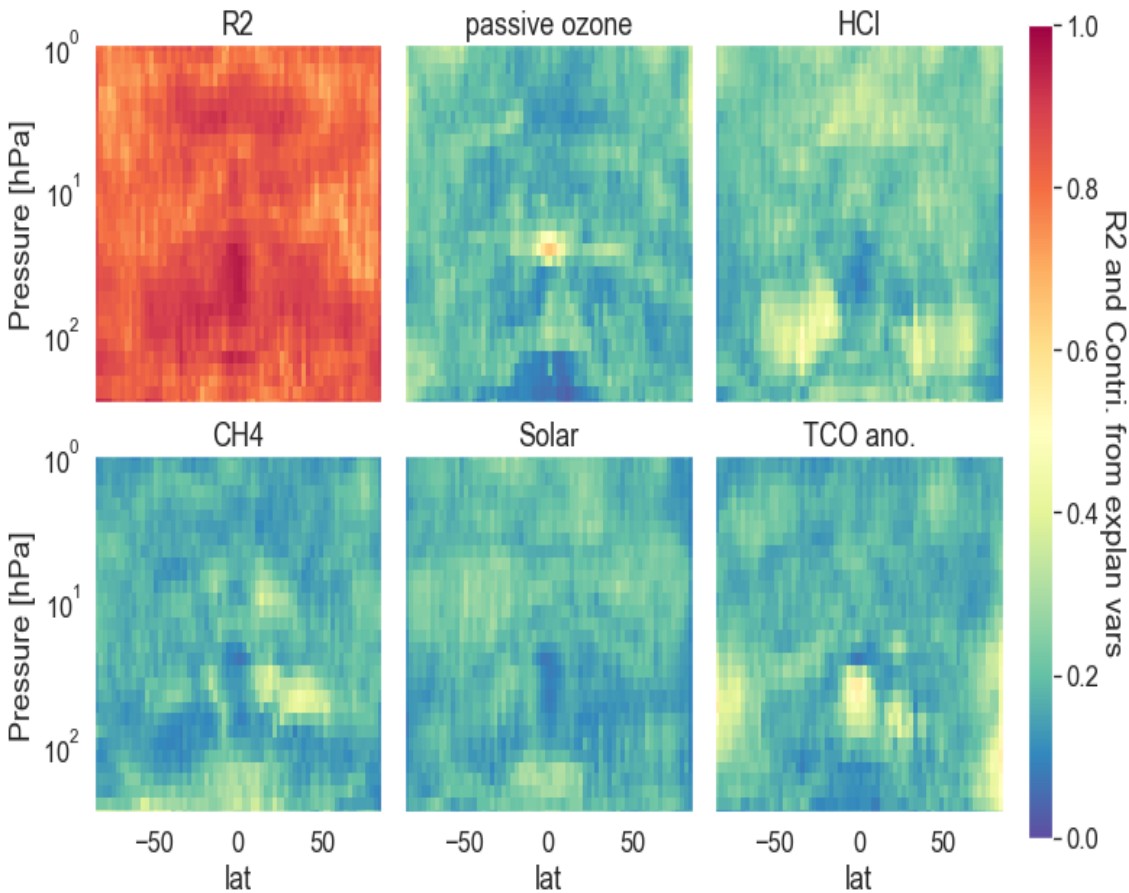

**Figure 2.** Latitude-pressure cross sections of the variance ($R^2$) and regression coefficients from passive ozone, $HCl$, $CH_4$, solar and total column ozone anomaly (see main text equation 1).

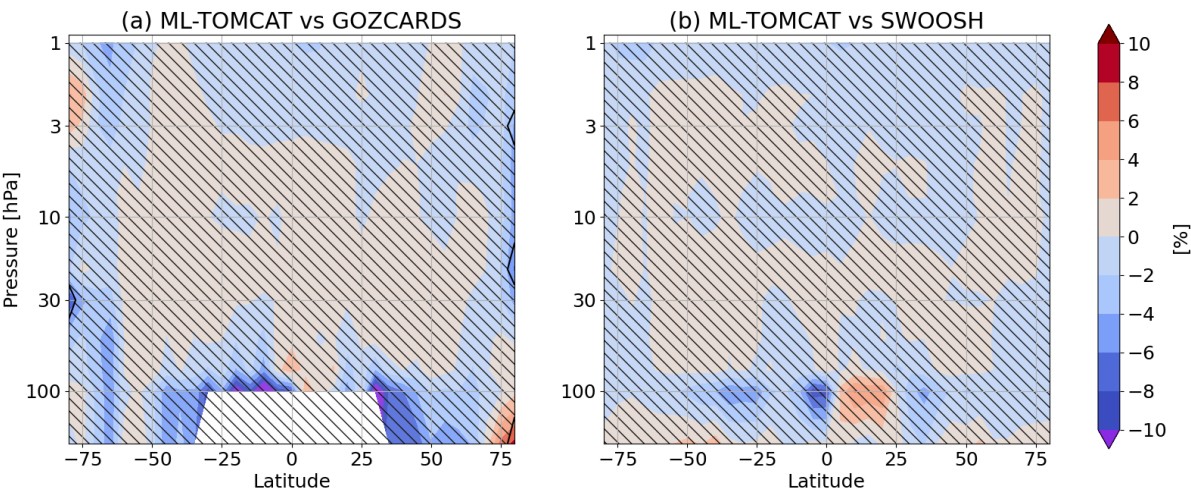

**Figure 3.** Relative differences (in percent) as a function of pressure and latitude between ML-TOMCAT and (a) GOZCARDS V2 (Froidevaux et al., 2019) and (b) SWOOSH (Davis et al., 2016). Stippling indicates regions where differences are smaller than one standard deviation.

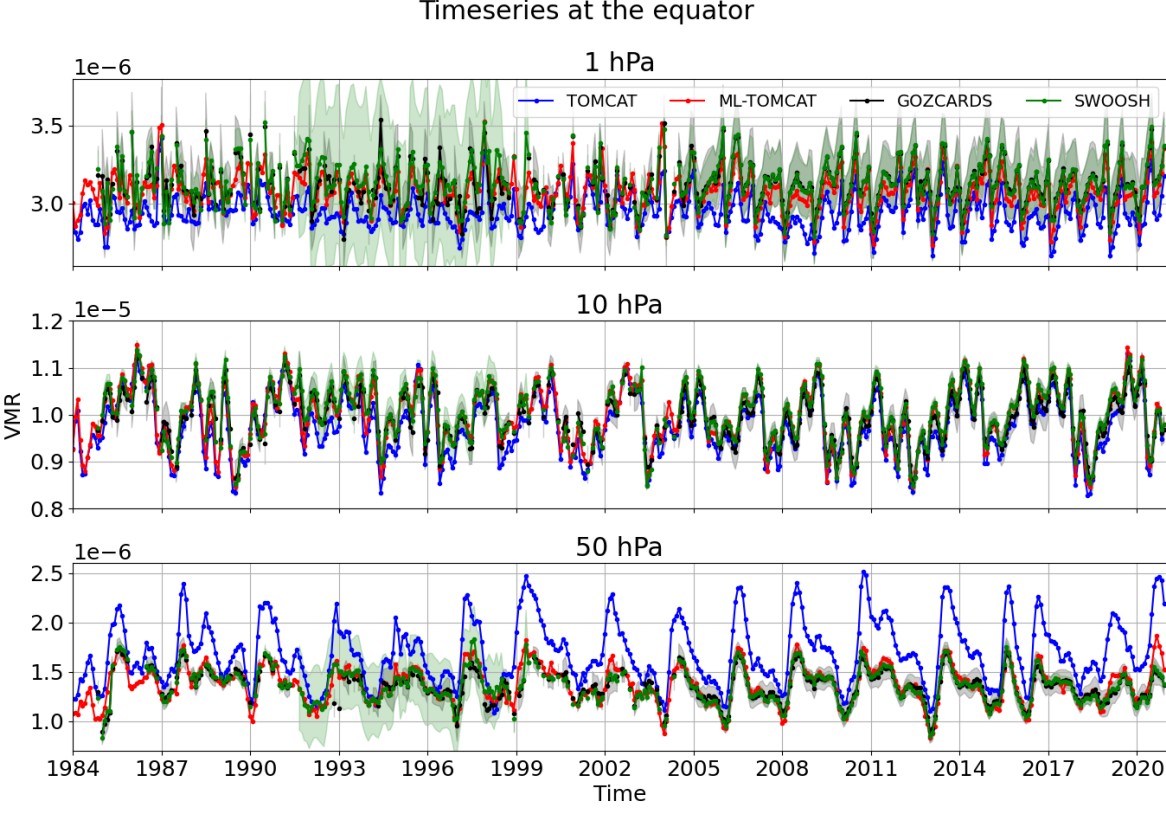

**Figure 4.** Comparison between TOMCAT (blue lines) and ML-TOMCAT (red lines) ozone mixing ratios over the equator ($0°$) at (a, top) 1 hPa, (b, middle) 10 hPa and (c, bottom) 50 hPa. Satellite-based ozone mixing ratios from GOZCARDS (Froidevaux et al., 2019) and SWOOSH (Davis et al., 2016) data sets along with their uncertainty estimates (shaded) are shown with black and green coloured lines, respectively.

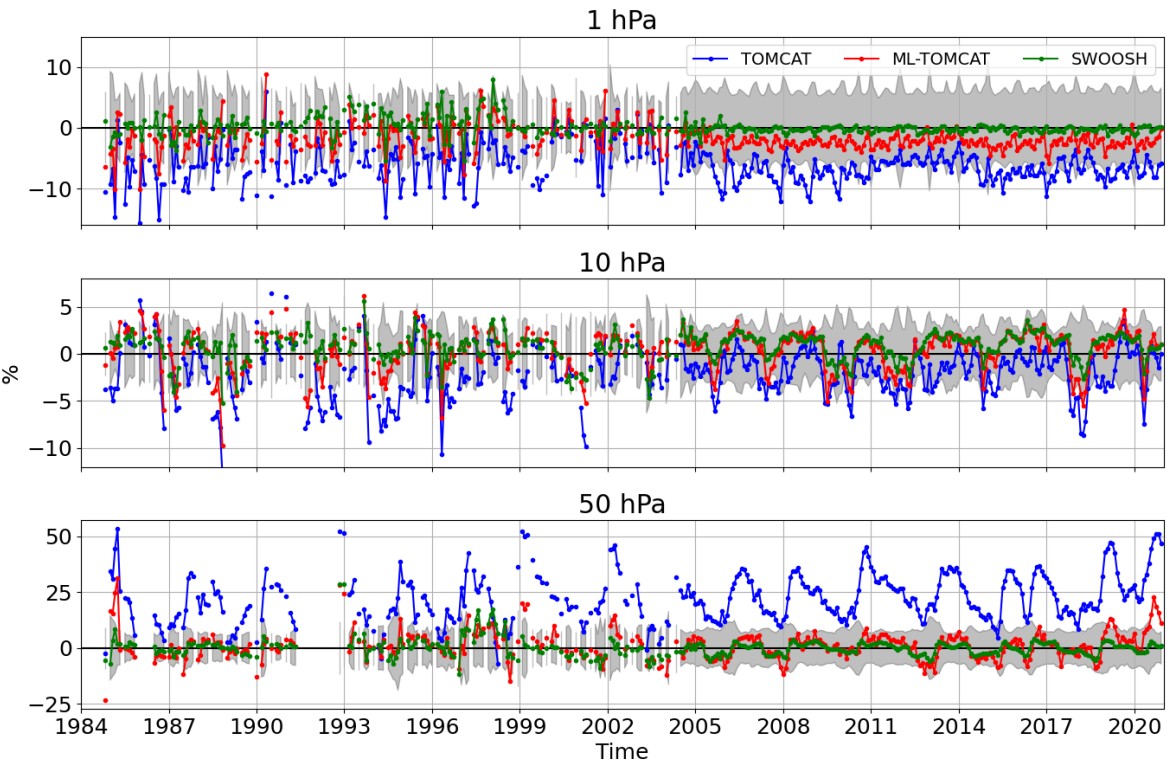

**Figure 5.** As Figure 4 but for the residuals, i.e. relative differences between SWOOSH (green), TOMCAT (blue) and ML-TOMCAT (red) ozone with respect to GOZCARDS ozone.

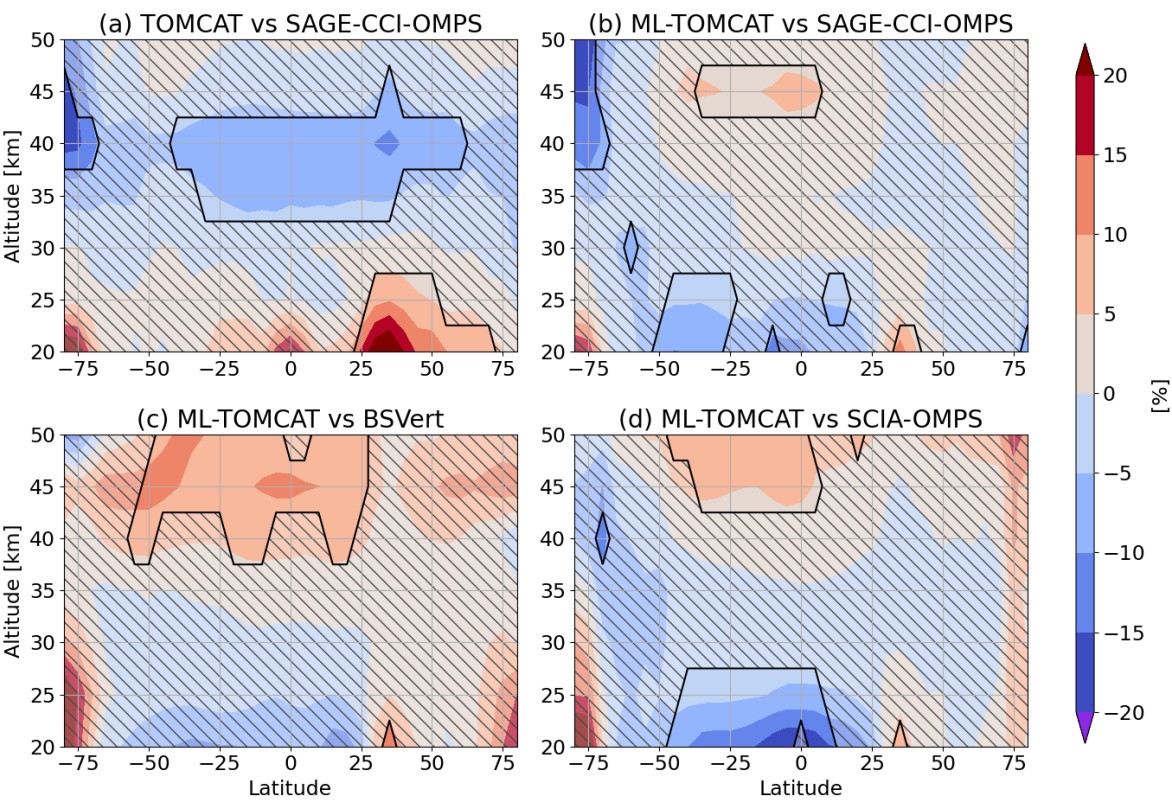

**Figure 6.** Relative difference (%) as a function of latitude and altitude between (a) TOMCAT versus SAGE-CCI-OMPS (1985-2019) and ML-TOMCAT versus (b) SAGE-CCI-OMPS (1985-2019), (c) BSVert (1985-2017) and (d) SCIA-OMPS (2002-2019), averaged over the respective time series. Stippling indicates regions where differences are smaller than one standard deviation.

**Figure 7.** Comparison between TOMCAT (blue lines) and ML-TOMCAT (red lines) ozone mixing ratios over the equator (0°) at (a, top) 45 km, (b, middle) 35 km and (c, bottom) 25 km. Satellite-based ozone mixing ratios from SAGE–CCI–OMPS, BSVert (Hassler et al., 2018a) and SCIA-OMPS (Arosio et al., 2019) data sets are shown with black, green and cyan-coloured lines, respectively.

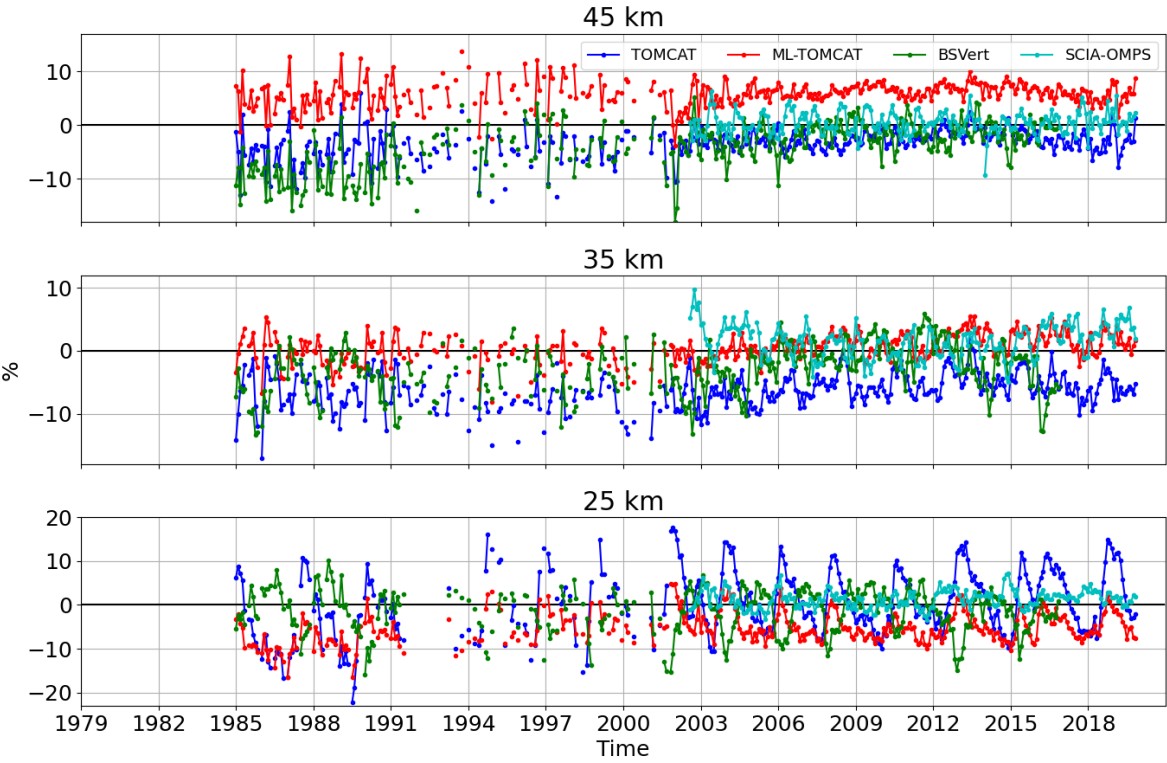

**Figure 8.** Same as Figure 7 but for the residuals, i.e. relative differences between TOMCAT (blue), ML-TOMCAT (red), BSVert (green) and SCIA-OMPS (cyan) ozone with respect to SAGE–CCI–OMPS.

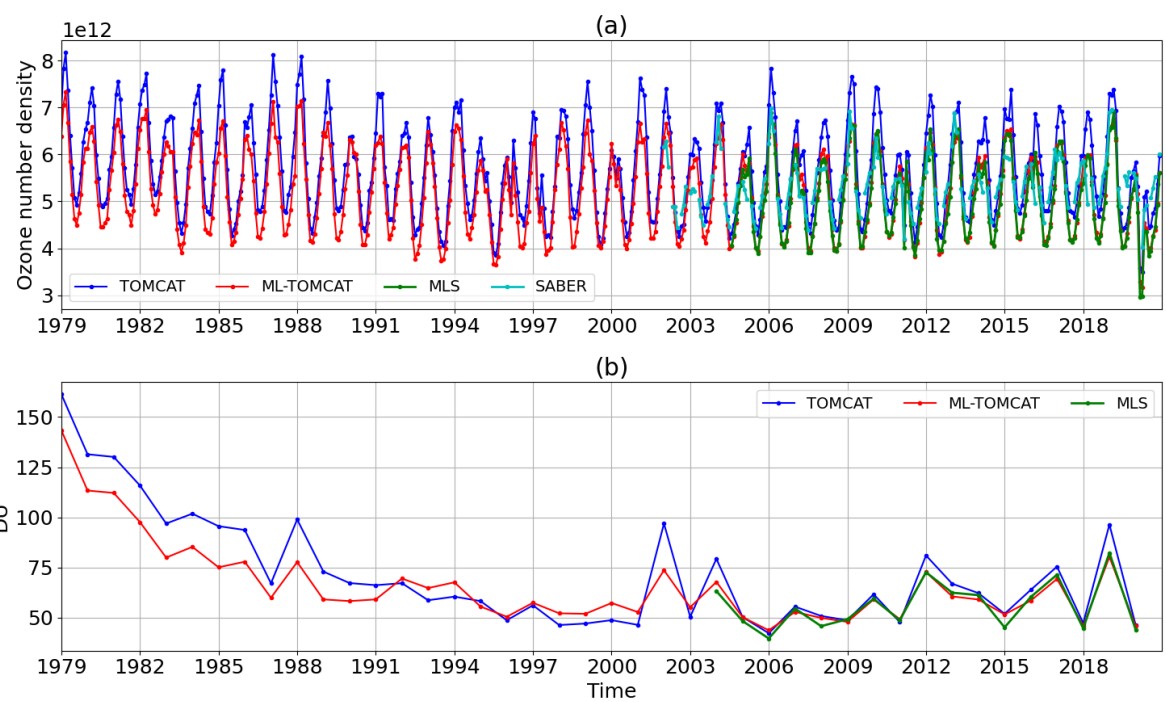

**Figure 9.** (a) Ozone concentration time series (molecules cm$^{-3}$) at 18 km over the Arctic region (latitudes poleward of 70°N). Aura-MLS and the Sounding of the Atmosphere using Broadband Emission Radiometry (SABER, Rong et al., 2008) data are superimposed on TOMCAT and ML-TOMCAT time series. (b) Mean ozone sub-column (DU) between 12-20 km for September and October each year over the Antarctic region (latitudes poleward of 70°S).

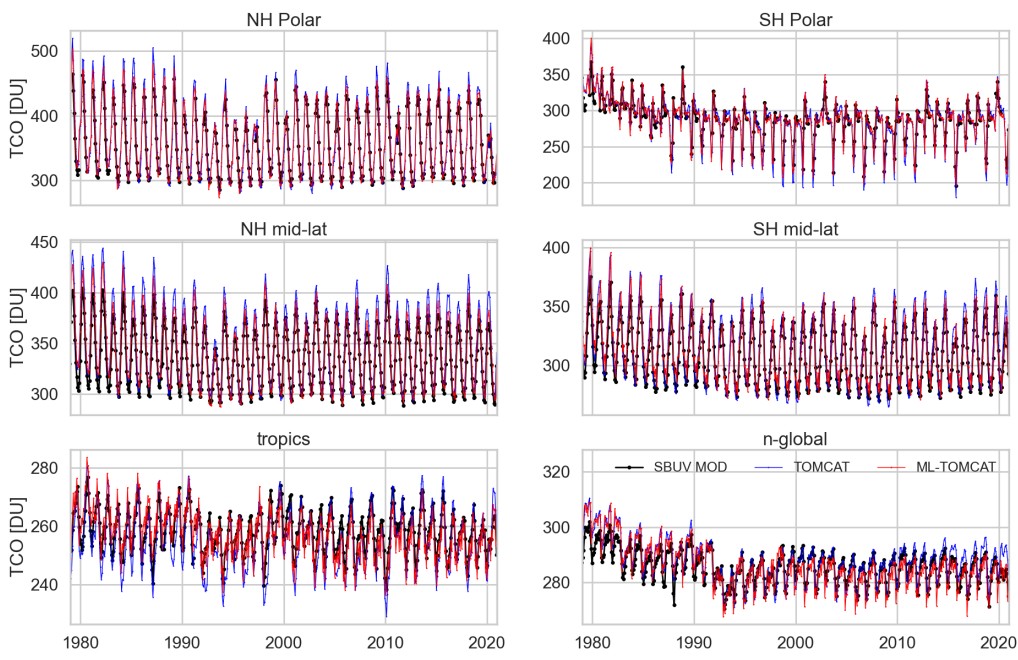

**Figure 10.** ML-TOMCAT (red line) and TOMCAT (blue line) total column ozone comparison with SBUV merged ozone data (MOD, black line) obtained from https://acd-ext.gsfc.nasa.gov/Data_services/merged/index.html. Monthly mean total column time series are shown for six latitude bins: Arctic (60°N-90°N), Antarctic (60°S-90°S), NH mid-latitudes (35°N-60°N), SH mid-latitudes (35°S-60°S), tropics (20°S-20°N), and near global (60°S-60°N).