# Peer review of "ML-TOMCAT: Machine-Learning-Based Satellite-Corrected Global Stratospheric Ozone Profile Data Set from a Chemical Transport Model"

_Earth System Science Data, 2021_

## Author Comment (AC1)

**Replies to Reviewer 1**

Review of "ML-TOMCAT: Machine-Learning-Based Satellite-Corrected Global Stratospheric Ozone Profile Dataset from a Chemical Transport Model"

**GENERAL COMMENTS**

First, I think that what this team has done here is excellent, very much needed, and that the data set arising from this work will be very useful. It is something I have always wanted to do myself but, lacking infinite time, have been unable to. What is presented in this paper is what I see as an ideal combination of our knowledge of stratospheric chemistry with all available stratospheric ozone profile measurements to create a definitive 42-year history of the vertical distribution of ozone in the stratosphere.

The only superior approach that I could think of would be a version of TOMCAT that has a 4Dvar assimilation of all available ozone profile measurements. You may want to say something about how your approach differs from what would be achieved from a data assimilating version of TOMCAT and why nobody has gone down the route of creating a data assimilating version of TOMCAT for this purpose.

I found that the description of exactly how the ML-TOMCAT database was created was too terse and I would like to see much more detail on that, i.e. sufficient for a reader to reliably replicate what was done.

**We thank the reviewer for his/her encouraging comments. We agree with the reviewer that some part of the manuscripts were a bit short. In the revised manuscript we expanded the methodology section. A simplified description about RF we added is**

"The RF is a supervised machine learning (ML) algorithm that uses an ensemble of decision trees (e.g. Breiman 2001, Svetnik et al., 2003). A decision tree can be considered as a flow chart used in computer programming (a tree-shaped schematic) that is generally used to show a statistical probability or path of action. A single decision tree in a RF can be considered as a random tree in a forest of decision trees. Each decision tree consists of three components: decision nodes, leaf nodes, and a root node. The root node and decision nodes of the decision tree represent the explanatory variables. The leaf node represents the final output. The explanatory variables used in our analysis are explained at end of this section.

A decision tree algorithm divides the data set into branches (using true and false criterion), which further segregate into other branches until a leaf node (or result node) is reached. Multiple trees are constructed by randomly sampling data points multiple times (e.g. bootstrap method).

We also added a paragraph to inform the readers about the usage of the data assimilation techniques to create ozone profile data sets, as

"Another widely accepted approach is using data assimilation techniques to create observational based data (e.g. Feng et al., 2008, Skachko et al., 2014, Errera et al., 2019, Wargan2020). However, for ozone only a few instruments such as MLS provide relatively long-term ozone profile measurement. For the pre-MLS time period, very few observations are available that could provide good constraint on the assimilation system. As the forward model is generally forced with available (re)analysis dynamical fields, reanalysis datasets are also prone to the inhomogeneities in the forcing fields along with any discrepancies in the chemical scheme."

**SPECIFIC COMMENTS**

Line 58: Rumour has it, but who knows whether it is true, that the true expansion for BDBP is "Birgit's Database of Biblical Proportions".

**Indeed, we heard the same. However, we assume that this is a tongue-in-cheek comment that does not require any change to the paper.**

Lines 95-96: Does this mean that the resultant ML-TOMCAT ozone profile database is not longitudinally resolved? That's a little disappointing. I thought that with TOMCAT being 3D you would be able to create a 3D ozone profile database.

**## Yes, for time being these are monthly mean zonal mean profiles. We are hoping to create 3D ozone profiles in our next version (MLTOMCAT V2) that might include tropospheric corrections.**

Section 3: I suspect that most readers of your paper will be like me, i.e. experts in stratospheric ozone with some (or maybe even little) expertise or knowledge of machine-learning methods. When I think of a 'random forest' I think of someone who has planted trees somewhere arbitrary to claim carbon credits. All I am saying here is that you may need to be more pedagogical in your approach to Section 3. You may need to describe in greater detail, and at a higher level, the machine-learning methods that you are using. Don't miss the opportunity to educate your reader to the level that they require to understand your paper, being cognizant of the level of knowledge your typical reader is likely to have. If you do it well, they will be eternally grateful. Test the material on non-ML-experts to see whether it has been pitched at the appropriate level. Just as one small example, on line 113 you refer to 'the model'. I suspect here that you are no longer referring to TOMCAT but now to some model underlying the random forest? Line 113 also refers to 'learners' for the first time without describing what these are. You will need to rewrite this section being extra careful to maintain clarity.

**## We agree with the reviewer. Initially we also had some confusion about Random Forest, hence now we included detailed information in the revised manuscript. Learners are replaced with "explanatory variables".**

Line 117: Isn't the full correct name for the package scikit-learn?

**## Both are the same. It is installed as scikit-learn but module is imported as sklearn**

Line 132: In the context of regression modelling, would a better term for 'learner' be 'predictor' or 'basis function' or 'explanatory variable'? I would suggest that, to the extent possible, you use terms that will be more familiar to the majority of your audience (atmospheric scientists) rather than terms used more by the data science community.

**## We agree. Learner has been replaced with "explanatory variable"**

Equation (1): shouldn't there be coefficients at the front of each of these terms as in a standard regression model, these being the regression model coefficients? You refer to these coefficients on line 191. Otherwise this equation doesn't make sense to me. It doesn't even make sense in terms of units, e.g., dO3 would be measured in ppm and dTCO in DU. OK, maybe you are let off the hook by the 'and so all the predictor time series are detrended and normalised between 0 to 1' clarification at the end.

**We tried different algorithms and some of them need standardisation (or scaling between 0 and 1) otherwise the regression coefficients varies over several order of magnitudes. Although for Random Forest it is not required, we decided to use normalised time series. Regression coefficients have been added in Eq. 1.**

Line 140: I assume that TCO fields are not available from ERA5?

**Yes, as discussed in methodology section, total column is from Copernicus Climate Change Service (C3S) total ozone data (1979--2018) and OMI for later two years.**

Line 163: And this is no small caveat. I think that it would be worth pointing readers to the SPARC S-RIP activity here.

**Thanks very good suggestion. We added a couple of sentences:**

"Therefore recently SPARC initiated the SPARC Reanalysis Intercomparison project (S-RIP) providing guidance to the future reanalysis activity. S-RIP also aims at performing a comprehensive evaluation and intercomparison of different reanalysis data products. For details see https://www.sparc-climate.org/sparc-report-no-10."

Figure 1: Why start in December and finish in November? Here, our year runs from January to December.

**Corrected.**

Line 169: You say '(2006-2020)' but the figure caption says 2001-2020.

**Corrected caption for Figure 1. It should be 2006-2020 (15 years) covering AURA/MLS sampling period that has much denser and better vertical sampling. In this period, SWOOSH is largely based on MLS data.**

I think that throughout the paper things would be clearer if you replaced 'the model' with 'TOMCAT' wherever that makes sense since there is also the regression model and you don't want the readers to constantly be having to figure out which one you are referring to when you say 'the model'.

**## We agree. Now we have clear separation between chemistry models /TOMCAT or the regression model.**

Line 175: I find this confusing. On line 170 you state that positive biases occur in the upper stratosphere and negative biases occur in the lower stratosphere. But then on line 175 you explain the source of negative ozone biases in the upper stratosphere (even though this is not what is observed) and, similarly, on line 180 you explain the source of positive biases in the lower stratosphere whereas what is observed is negative biases. Am I missing something here?

**We apologise for the confusion. As shown in Figure 1, compared to SWOOSH, TOMCAT ozone profiles show positive biases (TOMCAT values are larger) in the lower stratosphere and negative biases in the mid-upper stratosphere. To be consistent, we follow Bias = TOMCAT -SWOOSH definition. This means positive bias = TOMCAT values are larger, negative bias = SWOOSH values are larger.**

Line 195: And presumably these RF-derived biases extend to regions of the globe for which observed ozone profiles are not available?

**Yes, for missing SWOOSH data points, ozone difference values are linearly interpolated. At the poles, we assume TOMCAT has zero biases when there are no SWOOSH data points.**

Line 202: Here you refer to TOMCAT-SWOOSH differences whereas on line 170 you said 'SWOOSH minus TOMCAT'. I find this confusing.

**Sorry, corrected it as TOMCAT – SWOOSH differences (though here we are only discussing differences between SWOOSH and TOMCAT ozone profiles at high latitudes.**

Figure 2: Presumably the sum of all 5 fields other than the R^2 field should sum to the R^2 field? Has that been confirmed?

**Indeed that is true. We added a sentence in a revised manuscript.**

Line 209: I think that it would be worth saying more about why the HCI basis function has such high explanatory power in the tropical lower stratosphere.

**We are sorry but there was some error in the plotting routine that was found while replotting the graphs using different colour scale (Reviewer 2). However, we agree with the reviewer, hence we added some more discussion in a line that "HCI can be considered as both dynamical and chemical proxy. Upper stratospheric HCI is transported downwards via BD circulation (e.g Mahieu et al., 2014). So HCI variations in this region can be considered as a combination of changes in the ascending branch of the BD circulation as well as horizontal isentropic transport".**

Line 238: This suggests that in Figure 3 there should be stippling to show where the differences are statistically significantly different from zero at the 2 sigma level.

*## Done. We now also included stippling in Figure 6.*

Line 247: I assume you mean significant improvements compared to TOMCAT? You should then say so.

**## Done**

Line 252: Why the need to use SAGE II and HALOE when UARS MLS always covers the lower latitudes (34S to 34N)?

**Yes, that is correct. It should be the difference in spatial sampling, we removed the sentence that was suggesting otherwise.**

Line 262: You say with respect to GOZCARDS but in the title of the figure it says "w.r.t. SAGE-CCI-OMPS"?

**## Corrected**

Line 266: It wasn't clear to me what you meant by 'uneven variations'.

**Here we meant that there are some periods where TOMCAT -SWOOSH differences keep increasing or decreasing for 3 to 4 years. For example, from 2008 to 2014, TOMCAT biases w.r.t. SWOOSH show a monotonic decrease that might be linked to the inclusion of AMSU temperature profiles measurements in the re-analysis that have better vertical resolution compared to MSU profiles (ozone at this altitude show a compact correlation with temperature fields (e.g. Dhomse et al., 2016)).**

Line 275: You need to cite a paper to support this assertion that there is a bias in the ERA5 reanalyses in 2020.

**Manuscript is under preparation but we presented the analysis at the Quadrennial Ozone Symposium 2021 and is cited as Chrysanthou et al, QOS, 2021 in the revised manuscript.**

Lines 276-278: As it is currently written, this sounds like conjecture. You either need to do the analysis to investigate whether or not this is the case or cite something that gives some credence to the assertion.

**Now we revised it to say "it needs further investigation".**

Line 279: I would prefer to see this referred to as 'Comparison with ozone concentrations reported on altitude levels'. In fact I would prefer to see the word 'altitude' rather than 'height' used throughout the paper, unless you are specifically referring to geopotential height rather than geometric altitude in which case I think that you should always be explicit and say 'geopotential height'. After all, Figure 6 refers to altitude rather than height.

**Done**

Line 288: Yes, but if I remember correctly, the TOMCAT profiles are not used directly in the BSVert ozone data set but only as a transfer standard to calculate relative biases. So there is no reason why BSVert ozone cannot be biased against TOMCAT. I would suggest that you read Hassler et al., 2018 very carefully.

**Revised it to reaffirm that TOMCAT profiles are used only as transfer function.**

Line 289: This is not true. The negative biases extend to 25N.

**Revised it to say that biases are negative in the tropics and SH mid-latitudes and then turn positive in the NH (beyond 25 N).**

Line 310: Should this be 'ozone number density profile'?

**Yes. Corrected.**

**GRAMMAR AND TYPOGRAPHICAL ERRORS**

I have suggested some, but not all, grammar and typographical corrections. I am hoping/expecting that further corrections will be made by the author(s) and the ESSD editorial staff.

Line 3: 'Satellite instruments obtain stratospheric ozone profile measurements' sounds rather convoluted. Why not just say 'Satellite-based instruments measure stratospheric ozone profiles'.

*## We have rephrased as 'Satellite based instruments provide stratospheric ozone profile measurements'.*

Line 37: Replace 'which could variability in ozone trends' with 'which could induce variability in ozone trends'.

**## Done**

Line 40: Replace 'As there is no' with 'As there are no'.

**## Done**

Line 54: Replace 'data was' with 'data were'.

**## Done**

Line 65: replace 'merged data' with 'merged data set'. Likewise on line 72 replace '(SWOOSH) data' with '(SWOOSH) data set'.

**## Done**

Line 75: Replace 'One of major' with 'One of the major'.

**## Done**

Line 81: Replace 'firstly' with 'first'.

**Done**

Line 111: Would it not be clearer to replace 'algorithm by splitting observations' with 'algorithm to split observations'?

**As per reviewer's suggestion, the whole paragraph has been revised.**

Line 119: Replace 'on to' with 'onto'.

**Done**

Line 120: Replace 'data is' with 'data are'.

**Done**

Line 152: Isn't the IUPAC convention that it is 'sulfate' rather than 'sulphate' irrespective of which side of the Atlantic you're one?

**Indeed that is a bit confused. For the last few years, the stratospheric aerosol community adopted "sulfate" (e.g. https://www.sparc-climate.org/activities/stratospheric-sulfur/), so now we have replaced "sulphate" with sulfate".**

Line 198: Replace 'that RF regression model' with 'that the RF regression model'.

**## Done**

Line 216: Replace 'This means although' with 'This means that although'. *## Done.*

Line 226: Replace 'data was processed' with 'data were processed'.

**## Done**

Line 227: Replace 'geopotential' with 'geopotential height'.

**## Done**

Line 320: Replace 'is able to polar' with 'is able to model polar'.

*## we have rewritten as "is able to simulate polar".*

Lines 323-325: This sentence lacks a verb.

**## Done**

Line 350: I believe this should be 'Hassler et al., 2018'? And please fix the reference accordingly - you appear to have used first names rather than surnames.

**## Done**

---

## Author Comment (AC2)

Replies to Reviewer 2

Review of the manuscript "ML-TOMCAT: Machine-Learning-Based Satellite-Corrected Global Stratospheric Ozone Profile Dataset from a Chemical Transport Model" by S.S. Dhomse et al.'

The manuscript "ML-TOMCAT: Machine-Learning-Based Satellite-Corrected Global Stratospheric Ozone Profile Dataset from a Chemical Transport Model" by S.S. Dhomse describes a new stratospheric ozone profile dataset and the machine-learning method used for its creation. The basis for the new dataset are profiles from a simulation provided by the chemical transport model TOMCAT. Biases between these profiles and profiles from the well-established ozone profile dataset SWOOSH are then calculated and used as input for a Random Forest regression method with five learners (or basis functions). The obtained coefficients for these five learners are then used to bias correct the entire TOMCAT simulation. The resulting ML-TOMCAT dataset is then compared to several different available observational datasets and differences are discussed. The manuscript is very well written, well structured, and the topic lays within the scope of the ESSD journal. However, there are a few things that I think would help to improve the manuscript, and that I would suggest the authors to consider while revising the manuscript. Comments highlighting these issues are outlined below.

*## We thank the reviewer for his/her positive comments. Our replies are given below (in red italics).*

General comments:

As the authors state in the manuscript, merging techniques for ozone datasets can result in significant uncertainties within the created dataset, and can be a considerable source of differences between merged datasets. The applied machine-learning method to correct the biases between the TOMCAT simulation and SWOOSH is therefore one of the main parts of the manuscript. However, at the moment it feels too short and not detailed enough considering its importance for the creation of the dataset. Are the five learners applied in the same form to all latitudes and altitudes/pressures? How big are the residuals after the Random Forest regression has been applied? Why was the tropospheric part of the profile also put through the Random Forest regression although there is no data from SWOOSH and the ML-TOMCAT dataset is recommended only for stratospheric values?

It is not clear enough where the focus of the ML-TOMCAT dataset is placed. The title of the manuscript suggests that it would the stratosphere, and in the summary section the recommendation is given to only use the dataset between the tropopause and 0.1hPa, but the creation of the dataset is described for all pressure levels between 1000hPa and 0.1hPa. This is confusing. Please add more explanations why either the tropospheric levels are necessary for the creation of the dataset or why they were created but should not be used.

*## We agree with the reviewer that some of the description was too brief. In the revised manuscript we explain that data outside stratospheric levels is primarily based on TOMCAT chemistry. Only a crude correction is applied to TOMCAT ozone profiles outside the stratosphere primarily based on known biases. As the reviewer correctly pointed out, we have been explicit that ML-TOMCAT profiles should be used for stratospheric studies. Now we also explain that we plan to release a future version where tropospheric profiles would be corrected by using merged tropospheric ozone data products (e.g. Tropospheric Ozone Assessment Report (TOAR)). So, at this stage tropospheric and lower mesospheric levels can be considered as a placeholder for next version of the ML-TOMCAT profiles. This is explained at the end of the revised manuscript.*

*As for the residual, we are bit confused as Figures 5 and 8 do compare the residuals. A quick glance at Supplementary Figures also gives the overall idea about the residuals. We have now modified Figures 3 and 6 to have stippling to represent where differences between ML-TOMCAT and particular evaluation data set are statistically insignificant.*

More specific comments:

Line 18: Should be "…within uncertainties of the…" rather than "…within uncertainties in the…"?

*## Corrected*

Line 29: Should be "…similar or larger magnitude…" rather than "…similar magnitude or larger…"?

*## Corrected*

Line 37: Word missing? "…which could variability in ozone trends."

*## Corrected as "which could induce variability in ozone trends."*

Line 66: Remove "s" from "profiles"

*## Corrected*

Line 102: Add "the" between "…throughout troposphere."

*## Corrected*

Line 117: "Random Forest" has been introduced already and does not need to be written out here anymore.

*## Corrected*

Line 121: The selection of the years to train the machine-learning model (1991-1998 and 2005-2016) seems very random here and is not explained. It is explained later in the manuscript, but I think it would be good to add an explanation here as well (not just that it is a 20-yr period, but WHY exactly these 20-yrs were chosen).

*## Indeed, we should have been explicit about the training period. A clear statement concerns the MLS time period as it has much better vertical and temporal sampling. We have also added two more explanations: a) it includes the period with maximum chlorine loading and b) it sufficiently samples the observed ozone variability to go along with and c) only MLS instruments provide dense ozone profile measurements at good vertical resolution.*

Line 129: Please explain here what you mean with "passive ozone"

*## Corrected*

Line 169: Here it states that the climatology was calculated for the period 2006-2020, but the figure caption (Figure 1) states that the climatology was calculated for the period 2001-2020. Which is correct?

*## Corrected as 2006-2020.*

Line 201: the sentence part about positive and negative ozone biases is confusing here since it refers to Figure 1 and the paragraphs explains details of Figure 2. Maybe add a reference to Figure 1 at the end of the sentence?

*## Corrected. Now all the differences refers to SWOOSH – TOMCAT. Hence in the upper stratosphere they should be positive and in the lower stratosphere they are negative. See also replies to Reviewer 1.*

Line 209: What could be the reason for the significant HCl coefficients in the tropical lower stratosphere? You mention that they are present there but don't offer an explanation.

*## Corrected*

Line 225: Remove the comma after "Since, …"

*## Corrected*

Line 246: Reference to Figure 1 is not correct, I guess, since Figure 1 does not show ML-TOMCAT data.

*## Corrected, also note the revised colour scale (Reviewer #1)*

Line 321: ";" between the two references should probably be an "and"

*## Corrected*

Line 322: I guess you mean "North Polar region" instead of "North Pole"? It might be good to mention here as well which latitudes are covered by the North Polar regions.

*## Corrected as Arctic.*

Line 323: I think "South Polar region" is more appropriate than "South Pole".

*## Corrected as Antarctic*

Figure 2: The differences in the blue color range a very hard to distinguish. It might be helpful to change the color scale somewhat to help the reader understand the discussion about these differences from Section 4.2.

*## Corrected*

Figure 5: The title says "w.r.t. SAGE-CCI-OMPS", but it should be "w.r.t. GOZCARDS"

*## Corrected*

Figure 9: Last line of the figure caption: the ")" is missing after "70°S

*## Corrected*

---

## Author Response (AR2)

**Replies to the Editor:**

**Comments to the authors**

Thank you for the thorough revision of your manuscript. I am happy to accept this paper if you address the two following minor issues:

1) "Stippling" in the figures, and in particular Figure 3: first, please abandon the term "statistical significance" (see https://www.nature.com/articles/d41586-019-00857-9 and the more detailed articles in the literature). The implication of "significance" and of the stippling in the plots is that the results are not meaningful if they are "not significant", but this is incorrect. I am not against stippling per se, although I much prefer to use two different saturation levels as this is psychologically much more intuitive and does not imply that you wish to discard stippled features. Please take time to think this through thoroughly, because it is very important for all scientific work.

**We are thankful to the Editor for raising this important aspect of manuscript writing. We agree with the suggestion and in the further revised manuscript we no longer use the term "statistically significant" anywhere. Now captions for Figures 3 and 6 clearly state that "stippling indicates the regions where differences are smaller than 1-sigma variability".**

2) I would like to see a better justification for including the tropospheric levels in the data product, even if they are almost exclusively TOMCAT simulation based. Please look at this from a user perspective: how do you envision your data product to be used? Is it for example important that the total ozone column is meaningful? (This means that stitching together ozone fields from two different product typically needs some bias correction). I think it is good to have the whole atmosphere covered, but I would like to be convinced more strongly.

**We agree with the Editor that some better explanation was needed for the including ozone profiles at all the levels. In the Methodology section we have clarified that the TOMCAT tropospheric values are based on a climatology (Logan et al., 1999) in the stratospheric model, and that these values are then used unchanged in ML-TOMCAT below 316 hPa. Updated files are uploaded on Zenodo and both the versions can be accessed via: https://zenodo.org/record/5651194**

We added an additional section (Section 4.3.4) to discuss total column comparisons and show that having whole atmosphere profiles helps to improve total column calculation. We have now added an additional figure in the manuscript (Figure 10) comparing total column ozone for six latitude bands (Arctic (60N-90N), Antarctic (60S-90S), NH mid-latitudes (35N-60N), SH mid-latitudes (35S-60S), tropics (20S-20N) and near global (60S-60N)) with SBUV merged data. The comparison clearly shows that ML-TOMCAT total column data shows much better agreement with the SBUV data compared to TOMCAT data. We make it clear that we include the tropospheric values to allow column comparisons such as this, but the tropospheric values should not be used for tropospheric studies. We also added a supplementary figure (Figure S22) to compare tropospheric ozone columns from ML-TOMCAT and TOMCAT. The TOMCAT data set shows a near-identical annually repeating cycle whereas tropospheric ML-TOMCAT shows variability with some short-term and long-term variations at different latitude bands. A further figure (S23) compares the TOMCAT and ML-TOMCAT mean latitude-altitude ozone cross section.